# RNA-Binding S1 Domain in Bacterial, Archaeal and Eukaryotic Proteins as One of the Evolutionary Markers of Symbiogenesis

**DOI:** 10.3390/ijms252313057

**Published:** 2024-12-04

**Authors:** Evgenia I. Deryusheva, Andrey V. Machulin, Alexey A. Surin, Sergey V. Kravchenko, Alexey K. Surin, Oxana V. Galzitskaya

**Affiliations:** 1Institute for Biological Instrumentation, Federal Research Center “Pushchino Scientific Center for Biological Research of Russian Academy of Science”, Russian Academy of Science, 142290 Pushchino, Russia; evgenia.deryusheva@gmail.com; 2Skryabin Institute of Biochemistry and Physiology of Microorganisms, Federal Research Center “Pushchino Scientific Center for Biological Research of Russian Academy of Science”, Russian Academy of Science, 142290 Pushchino, Russia; and.machul@gmail.com; 3Faculty of Informatics and Computer Engineering, MIREA—Russian Technological University, 119454 Moscow, Russia; alexey_junior02@mail.ru; 4Institute of Environmental and Agricultural Biology (X-BIO), Tyumen State University, 625003 Tyumen, Russia; svkraft@yandex.ru (S.V.K.); alan@vega.protres.ru (A.K.S.); 5Institute of Protein Research, Russian Academy of Sciences, 142290 Pushchino, Russia; 6Branch of the Shemyakin–Ovchinnikov Institute of Bioorganic Chemistry, Russian Academy of Sciences, 142290 Pushchino, Russia; 7State Research Center for Applied Microbiology and Biotechnology, 142279 Obolensk, Russia; 8Institute for Theoretical and Experimental Biophysics, Russian Academy of Sciences, 142290 Pushchino, Russia; 9Gamaleya Research Centre of Epidemiology and Microbiology, 123098 Moscow, Russia

**Keywords:** RNA-binding S1 domain, structural repeats, symbiogenesis

## Abstract

The RNA-binding S1 domain is a β-barrel with a highly conserved RNA-binding site on its surface. This domain is an important part of the structures of different bacterial, archaeal, and eukaryotic proteins. A distinctive feature of the S1 domain is multiple presences (structural repeats) in proteins and protein complexes. Here, we have analyzed all available protein sequences in the UniProt database to obtain data on the distribution of bacterial, eukaryotic and archaeal proteins containing the S1 domain. Mainly, the S1 domain is found in bacterial proteins with the number of domains varying from one to eight. Eukaryotic proteins contain from one to fifteen S1 domains, while in archaeal proteins, only one S1 domain is identified. Analysis of eukaryotic proteins containing S1 domains revealed a group of chloroplast S1 ribosomal proteins (ChRpS1) with characteristic properties of bacterial S1 ribosomal proteins (RpS1) from the Cyanobacteria. Also, in a separate group, chloroplast and mitochondrial elongation factor Ts containing two S1 structural domains were assigned. For mitochondrial elongation factor Ts, the features of S1 in comparison with the RpS1 from Cyanobacteria phylum and the Alphaproteobacteria class were revealed. The data obtained allow us to consider the S1 domain as one of the evolutionary markers of the symbiogenesis of bacterial and eukaryotic organisms.

## 1. Introduction

To date, in the Structural Classification of Proteins (SCOP) database, the β-barrel consisting of five antiparallel β-strands and one α-helix is classified as a separate group in the class of β-proteins. This structural fold was called OB-fold (oligonucleotide/oligosaccharide-binding fold). It was originally identified in a group of bacterial and yeast proteins as an oligonucleotide/oligosaccharide-binding domain [1]. Later, it was shown that proteins with OB-fold are able to bind various types of DNA, RNA and protein molecules [2]. In various proteins, the OB-fold size can vary from 70 to 150 amino acid residues, while the 3D structure (β-barrel) remains highly conserved. The loops connecting the β-strands vary in size and conformation, determining the specificity of protein binding. A distinguishing feature of this fold is the presence of its multiple copies in proteins and multiprotein complexes. Such multiple copying of the structure increases the affinity and/or specificity of protein binding to nucleic acid molecules. OB-fold is found in bacterial, eukaryotic, and archaeal proteins. The conservatism of the fold and the topology of the RNA-binding center on its surface make it possible to consider it as one of the “oldest” protein folds, tolerant to mutations and capable of accommodating the binding of a wide range of ligands.

Among all proteins containing OB-fold, the most numerous is the superfamily of nucleic acid-binding proteins (nucleic acid-binding proteins superfamily), in which the OB-fold is represented by the S1 domain [3]. Compared to other variants of the OB-folding (for example, the cold shock domain (CSD)), the S1 domain has the largest number of multiple copies (domains), depending on the taxonomic affiliation. For example, bacterial initiation factor IF1, eukaryotic translation initiation factor eIF2a, RNase E, transcription factor NusA, and some other proteins from lower and higher organisms are known to have one copy of the S1 domain [4,5]. However, there are proteins containing nine to twelve copies of the S1 domain. These proteins are found mainly in eukaryotes. For example, the yeast protein Rrp5p involved in the processing of pre-rRNA maturation of 40S and 60S ribosomal subunits consists of 12 repeated S1 domains [6].

One of the most interesting protein families containing the S1 domain is the S1 ribosomal protein family (RpS1). Multifunctional RpS1 is part of the 30S subunit of the ribosome and plays an important role in the initiation of mRNA translation, participates in elongation, and also performs a number of extraribosomal functions [7,8,9]. The RpS1 family (1324 sequences) accounts for approximately 20% of all bacterial proteins containing the S1 domain [3]. A distinctive feature of this family is the multiple copying of structural domains in bacteria. We have shown that only 53% of RpS1 proteins can be considered ordered proteins. The remaining proteins are characterized by the state of a molten globule. In addition, RpS1 is characterized by an equal ratio of the secondary structure of individual domains, which indicates a similar structural organization of individual S1 domains and multidomain S1 proteins, with relatively short flexible or disordered regions predominating in multidomain proteins. The lowest percentage of flexibility is characteristic of the central parts of multidomain proteins, which are apparently associated with the functionality of S1. The more stable and compact central part in multidomain proteins is important for interaction with RNA, while terminal domains are important for other functions [10]. Our analysis of the available structures of the S1 domains revealed that the most flexible region of the loop (40–50 amino acid residues, a.a.r.) in the domain could potentially be involved in interactions with natural ligands. Our results also showed that the number of possible functions of eukaryotic proteins is increased when increasing the number of structural domains and flexible linkers between domains, rather than by changing the characteristics of a single structural domain [11]. In addition, we have shown (http://oka.protres.ru:4200, accessed on 4 September 2024) that the number of domains in RpS1 is a distinctive characteristic for the phylogenetic classification of bacteria according to the main phyla [3,12] (Figure 1).

RpS1, containing a variable number of structural domains, is identified in 25 different bacterial phyla. At the same time, a high percentage of amino acid sequence identity is revealed between individual S1 domains of three and four domains containing proteins, which are represented by evolutionarily related phyla Cyanobacteria and Chloroflexia, as well as two and four domains containing proteins (phyla Actinobacteria and Chloroflexia). The identity of six domains containing RpS1 with individual domains of three, four and five domains containing RpS1 corresponds to the evolutionary relationships in the combined PVC group, which includes Planctomycetes, Verrucomycetes and Chlamydia phyla [12,13].

Thus, the study of the S1 domain from various eukaryotic, archaeal and bacterial proteins is of great interest from the point of view of the theory of symbiogenesis [14,15], according to which chloroplasts and plastids (organelles of higher plants, algae) appeared as a result of symbiosis of Cyanobacteria with the ancestor of archeplastids [16]. Although a number of ribosomal proteins have been identified in plastids, the molecular mechanisms that regulate chloroplast biogenesis remain not fully explained. Note that it was shown in [17] that the rice *ASL4* gene encoding RpS1 is required for chloroplast ribosome biogenesis and early development of chloroplasts. In addition, based on the similarity of S1 domains, it seems possible to confirm the theory of the evolutionary relationship of mitochondria in eukaryotic cells and microorganisms of the Alphaproteobacteria class [18,19].

## 2. Results and Discussion

### 2.1. Distribution of the S1 Domains in Eukaryotic, Archaeal and Bacterial Proteins

Mainly, the S1 domain is found in bacterial proteins (24142 sequences), and the number of domains varies from one to eight. Eukaryotic proteins (3263 sequences) contain from one to fifteen S1 domains, while in archaeal proteins (1162 sequences), only one S1 domain is always identified (Table 1, Appendix A).

The minimal size of an archaeal protein containing the S1 domain is 79 a.a.r. (part of the protein translation initiation factor IF-2 (aIF) subunit alpha (D2EE74)), and the maximum size is 898 a.a.r. (Archaea-specific RecJ-like exonuclease, contains DnaJ-type Zn finger domain protein (V4XD76)) (Table 1). The heterotrimeric archaeal translation initiation factor (aIF) plays a key role in the initiation of protein biosynthesis by delivering the initiator methionyl-tRNA to the P-site of the small ribosomal subunit [20]. The archaeal RecJ protein interacts with several proteins of the DNA replication apparatus, including helicase [21]. Also, the S1 domain is identified in the archaeal DNA-dependent RNA polymerase, which synthesizes RNA on a DNA template [22,23]. The S1 domain is part of the archaeal proteins that form the intracellular exosome protein complex involved in RNA processing. Thus, the noncatalytic components of the exosome (Rrp4 and Csl4), which form the upper part of the central channel of the circular exosomal structure, promote the interaction of the exosome with RNA and increase the efficiency of RNA binding and its degradation [24]. In addition, among the archaeal proteins, we found 66 sequences (5.6%) identified as proteins containing the RNA-binding S1 domain (protein size from 113 to 735 a.a.r.).

Among bacterial proteins containing the S1 domain, the most common are polynucleotide phosphorylase (PNPase), ribonucleases (RNase R, RNase E, RNase G), the NusA transcription factor and the RpS1 family. RNaseE and PNPase are part of the bacterial complex of the degradosome (analogous to the exosome) [25]. In general, almost all bacterial proteins containing the S1 domain have been shown to be involved in processes associated with interaction with various types of RNA. For some bacterial proteins, the number of structural S1 domains may differ for the same protein for different organisms. More detailed studies have shown that there is a correlation between the number of S1 domains in RpS1 in bacteria and their belonging to a specific phylum [12,13,26,27].

Such eukaryotic proteins as the catalytic components of the exosome Rrp44, Dis3, Csl4, RNA helicase, RNase E and RNase R, which bind RNA and orient it in an orthogonal direction to the exosome nucleus [28], also contain an S1 domain in their structure. The eukaryotic protein Rrp5, which is an important factor in the process of ribosome biogenesis, contains from one to fourteen S1 domains [3,29]. A large proportion and variability in the number of repeats in eukaryotic proteins is apparently associated with the necessity for individual proteins and proteins included in protein complexes to increase the affinity and specificity when binding various ligands.

Recently, it was proposed to single out archaea, which are most closely related to eukaryotes, as a separate group of organisms (Asgardarchaeota). In this group, eukaryotic signature proteins include proteins involved in the transfer, formation and transport of membrane vesicles and the formation of the cytoskeleton, as well as the ubiquitin system [30,31] and RNA polymerase, which is considered the most conserved and universal protein [32,33]. As noted above, the main function of the S1 domain is RNA binding; therefore, it is mainly included in the composition of proteins involved in different stages of the biosynthesis process. However, one copy of the S1 domain has been identified in archaeal proteins with an RNA polymerase signature (DNA-directed RNA polymerase subunit E, DNA-dependent RNA polymerase subunit E RpoE, etc.) and eukaryotic proteins (RNA polymerase II subunit), which appears to reflect horizontal gene transfer (HGT) [31]. In bacterial proteins, the RNA polymerase profile has not been identified (Appendix A), which requires further research to hypothesize about the “heredity” of this protein. Alignment of S1 domain sequences (Clustal Omega) from some proteins belonging to different domains of life (alignment 1 from Appendix A) revealed its lower percentage of homology (from 10 to 35%). This correlates with our data that the 3D fold of the S1 domain is more conserved than the primary structure [11].

In bacteria and eukaryotes, proteins containing only one S1 domain prevail. For this reason, it cannot be argued that all eukaryotic proteins with multiple S1 domains are of bacterial rather than archaeal origin. In addition, comparative genomics data suggest that the separation of eukaryotes into a separate domain was the result of the union of two cells, archaeal and bacterial, and this event is associated with the acquisition of mitochondria (“early mitochondrial” scenario) [34,35]. Thus, the number of S1 domains in eukaryotic proteins can be considered a consequence of gene transfer/inheritance from archaea or bacteria, or their common ancestor.

Among eukaryotic proteins, we found 630 sequences (19.2%) identified as proteins containing an RNA-binding S1 domain (protein size from 109 to 2812 a.a.r.) and also containing from one to fourteen S1 domains. At the same time, 19 sequences correspond to the 30S chloroplast ribosomal protein S1 (30S ribosomal protein S1, chloroplastic), and 2 sequences correspond to the mitochondrial ribosomal protein S1 (mitochondrial ribosomal protein S1). In the group of eukaryotic proteins containing two S1 domains, a group of chloroplast and mitochondrial Ts elongation factors (29 sequences) are also distinguished (Table 2).

It is known that eukaryotic genes, presumably of archaeal origin, primarily encode proteins involved in information processing (translation, transcription, replication, repair), while genes of putative bacterial origin encode mainly proteins with “operational” functions, such as metabolic enzymes, membrane components, and other cellular structures [36,37,38,39]. According to the data obtained, the S1 domain is a part of both proteins involved in both information processing (for example, archaeal translational factors, bacterial Transcriptional accessory protein Tex and eukaryotic RNA helicase) and “operational” proteins (archaeal RecJ-like, bacterial PNPase, eukaryotic Rrp44 and Dis3); therefore, it is impossible to speak of a clear heredity of this domain from archaea or bacteria to the eukaryotic genome.

Note that some proteins (bacterial, archaeal and eukaryotic) containing the S1 domain are identified as uncharacterized proteins, the systematic study of which in the future can reduce the gaps in their annotation, and also become the basis for future fundamentally new studies [40] (Table 1, Appendix A).

### 2.2. Structural S1 Domain Repeats

The presence of structural repeats in proteins is usually considered a successful evolutionary strategy, since the regularity of the fold structure and the diversity of the three-dimensional assembly leads to the existence of molecules of various sizes with many significant functions [41]. A large number of protein structures containing repetitive structural elements has led to the creation of several approaches to their classification to further understand the relationship between the protein sequence, their structure and function, as well as the evolutionary mechanisms of their development [41,42]. The S1 domain can be regarded as unique, since, with a highly conserved three-dimensional fold, within a single protein family such as RpS1, it may not have a high degree of identity [12]. Our studies [10,11,42] suggest that the common structure of proteins with such repeats is a bead-on-a-string molecule, in which individual beads correspond to globular S1 domains (Figure 1).

In general, the distribution of the S1 domain number in bacterial, eukaryotic and archaeal proteins correlates with the revealed features of other structural repeats. So, as mentioned in [43], eukaryotic proteins, as a rule, have more structural repeats than archaea and bacteria. For the S1 domain, the maximum number of repeats found is 15, identified in the eukaryotic protein (hypothetical protein (Q4S7N9)) (Table 1). Also, according to the data obtained, there is a positive correlation between the size of the protein sequence and the number of structural repeats. This is consistent with the fact that structural repeats are often involved in gene regulation and signaling [44,45]; namely, more complex organisms require more repeats to perform more functions. It is also seen that the S1 domain (along with the predominant number of proteins containing 1 S1 domain) in bacterial proteins prevails in the amount of 4 and 6, while in eukaryotic proteins, the most common are proteins containing 2, 3, 11 and 12 S1 domains. These data are also consistent with the results in [46], where it was shown that repeats are the result of duplication of several domains simultaneously. In addition, duplications are mainly found in the middle of the protein chain between other repeats [46], which was confirmed by us when studying the identity of neighboring S1 domains in the bacterial RpS1 [12].

### 2.3. Chloroplast Ribosomal Protein S1

Previously, we showed that the number of structural S1 domains in RpS1 can be considered as one of the features for the classification of the main bacterial phyla [12]. The alignment of three domains containing chloroplast ribosomal protein S1 (ChRpS1) (Table 2) revealed the identity of their sequences, equal to 51% (alignment 2 from Appendix A). Comparison with the full-size annotated sequences of RpS1 from the Cyanobacteria (http://oka.protres.ru:4200, accessed on 1 October 2024) showed that the average percentage of identity between them is 45%, which is quite high compared to the data obtained earlier in [12]. Note that all RpS1 from the Cyanobacteria contain three structural S1 domains, which is a distinctive feature of this particular division [12]. The most identical to each other are the third domains of RpS1 from the Cyanobacteria and three domains containing ChRpS1 (61%). The results obtained are consistent with our data on the identity of individual S1 domains (three domain-containing proteins) in the RpS1 [12], where it was found that the third S1 domain is the most conservative. Thus, for the first time, we have identified the ChRpS1 group with the characteristic properties of RpS1 from the Cyanobacteria (alignment 3 from Appendix A). 

Alignment of mitochondrial ribosomal proteins S1 (MtRpS1) with individual domains of six domain-containing RpS1 of the Alphaproteobacteria class showed the highest percentage of sequence identity (individual domains) found with the third and fourth domains of six domain-containing proteins (32%) (Figure 2). Comparison with other domains (first, second, fifth and sixth) of Alphaproteobacteria, as well as with various domains of bacteria from other phyla, gives a small percentage of identity. The results obtained correspond to our data on the identity of individual S1 domains (six domains) in the RpS1 [12], where it was found that the third and fourth S1 domains are the most conserved in the class of Alphaproteobacteria. 

Construction of the phylogenetic tree showed that individual S1 domains of Cyanobacteria are evolutionarily close to the S1 domains of the found chloroplast eukaryotic proteins (Figure 3A). According to the constructed tree, the most ancient organisms in this group are Cyanobacteria *S. elongates* (Uniprot ID: O33698), *G. kilaueensis* (Uniprot ID: U5QID8) and *Synechococcus* sp. (strain ATCC 27144/PCC 6301/SAUG 1402/1) (Uniprot ID: P46228). Note that the chronograms showed extremely polyphyletic relationships in Synechococcus, which has not been observed in any other Cyanobacteria [47,48]. In addition, the prevalence of HGT in Synechococcus lineages implies an excellent ability to receive and utilize exogenous DNA, putatively providing a selective advantage [47].

### 2.4. Elongation Factor Ts

In bacteria, elongation factors include EF-Tu, EF-Ts, EF-G and EF-P proteins; factors promote peptide synthesis in the ribosome at the stage of translation elongation and ensure continuous protein polymerization, starting from the moment when the initiator tRNA enters the P-site of the ribosome. The bacterial factor EF-Ts corresponds to the eukaryotic mitochondrial factor mtEF-Ts [49]. In this work, we have discovered a previously unstudied group of eukaryotic chloroplasts and mtEF-Ts. This group is characterized by the presence of two structural S1 domains in all cases (Table 2).

The first domains of chloroplast EF-Ts are 62% identical to each other, and the second domains are 45% identical. The first and second S1 domains of chloroplast elongation factor Ts are homologous by 40%. An analysis of the identity of S1 domains from the group of Ts chloroplast elongation factors with the bacterial RpS1 showed that the first domain of Ts chloroplast elongation factors corresponds to the third S1 domain of bacteria of the Cyanobacteria (52%) (Figure 2).

For mtEF-Ts, the alignment of the first domains gives a percentage of identity equal to 46%, and for the second, 60%. The first and second S1 domains of mitochondrial elongation factor Ts are homologous by 46% (alignment 4 from Appendix A). Analysis of the identity of S1 domains from the group of mitochondrial Ts elongation factors with RpS1 showed that the first domain of mitochondrial Ts elongation factors corresponds to the third S1 domain of Alphaproteobacteria (50%) (Figure 2). 

Construction of a phylogenetic tree showed that individual S1 domains of Alphaproteobacteria are evolutionarily very close to the S1 domains of the found mtEF-Ts (Figure 3B). According to the constructed tree, the most ancient organisms in this group are Alphaproteobacteria *A. lipoferum* (*strain 4B*) (Uniprot ID: G7Z834) and *R. prowazekii* (Uniprot ID: Q9ZD28).

It is known that structural analysis of some elongation factors shows that the eukaryotic mitochondrial factor folds into a three-dimensional structure similar to that observed in *E. coli* and *T. thermophilus* EF-Tu. The sequence of EF-Ts is less conserved than that of EF-Tu, and distinct schemes are observed for the interaction of EF-Tu and EF-Ts in different systems [50,51]. The three-dimensional models (obtained by AlphaFold) of the mitochondrial elongation factor Ts (Figure 4B) from *B. distachyon* (UniProt ID: I1HXX8 and chloroplastic elongation factor Ts (Figure 4A) from *A. thaliana* (UniProt ID: Q9SZD6) show that each has two highly conserved S1 domains located on the protein surface. As with bacterial elongation factors, these domains in eukaryotic factors play a role in RNA binding. 

Comparison with other domains of Alphaproteobacteria and Cyanobacteria, as well as with various domains of bacteria from other phyla (Actinobacteria, Betaproteobacteria, Chloroflexi, Bacteroids and others), gives a significantly lower percentage of identity. This confirms the relationship of the found groups of eukaryotic sequences with prokaryotic organisms of Alphaproteobacteria and Cyanobacteria.

### 2.5. S1 Domain as One Evolutionary Marker of Symbiogenesis

In [52], it was assumed that the early evolution of eukaryotes was not due to divergence but included the merging of evolutionary branches. According to this theory (the theory of symbiogenesis) [14,34,52,53,54], eukaryotic organelles, mitochondria and chloroplasts are former bacteria initially not related to eukaryotes (proteobacteria in the case of mitochondria and Cyanobacteria in the case of chloroplasts). These bacteria were engulfed by the eukaryotic cell and continued to live inside it, retaining their own genetic apparatus. Thus, the eukaryotic cell can be considered a multigenomic system. In addition, it is known that HGT is widespread in prokaryotes, due to which populations can quickly exchange parts of the gene pool [55]. Because of HGT, even in eukaryotes that have lost mitochondria, mitochondrial genes are initially found. One of the theories for the formation of internal symbionts is the assumption that the ancestors of mitochondria were intracellular parasites [56]. The authors of this hypothesis draw attention to the fact that in the group of Alphaproteobacteria, from which mitochondria originated, there are many intracellular parasites, for example, Rickettsia. The emergence of plastids, or chloroplasts, is usually considered a result of endosymbiotic absorption of Cyanobacteria (in plants, green and red algae), followed by distribution in other lines of eukaryotes through progressive cycles of secondary and even tertiary symbiogenesis [57] (Figure 5). Some proteins involved in key processes of organism functioning (cytochrome oxidases, ATP synthases, cytochromes, translation initiation and elongation factors and others) are considered markers of symbiogenesis [49,58].

As was mentioned above, the S1 domain, one of the structural variants of the OB-fold, is considered to be one of the “oldest” structural domains [5,59,60]. It is assumed that the variability of this fold allows the genome to retain the ability to organize multiple protein–DNA and protein–protein interactions necessary to maintain the functionality of organisms [61]. Previously, we have shown that the number of structural S1 domains in the RpS1 family can fit with the symbiotic model of the existence of bacteria from different taxonomic phyla. At the same time, more evolutionarily “ancient” phyla have a larger number of structural domains (mainly six), while more “young” ones have a smaller and more diverse number of structural S1 domains [13]. We found the shortest RpS1 sequence in parasitic bacteria (Mollicutes), which is characterized by a very small genome size (from 580 to 2200 kb). However, Mollicutes effectively perform all significant functions (including RNA binding by single-domain S1 ribosomal protein).

Based on the evolutionary relationships for different bacterial phyla, we have assumed that for RpS1 evolutionary development proceeded from multidomain proteins to proteins containing one S1 domain (reduction evolution). There is also evidence in [62] that mycoplasmas (single domain containing RpS1) are a regressive branch of the evolution of some Gram-positive bacteria or Firmicutes. It is more likely that the terminal S1 domains were “cut off” in the evolution and were saved [13].

As was shown, in the archaeal proteins, one S1 domain is always identified; therefore, it can be assumed that archaeal organisms do not need to multiply the S1 domain to increase affinity for RNA, which may be due to their habitat conditions. At the same time, the evolutionary reconstructions available in [63] suggest the presence of complex ancestors for most of the main groups of archaea, with subsequent evolutionary loss of genes, which is also consistent with the data obtained. 

It is known that eukaryotic genes of presumably archaeal origin encode, first of all, proteins involved in information processing (translation, transcription, replication and repair), while genes of presumed bacterial origin encode mainly proteins with “operational” functions, such as metabolic enzymes, membrane components and other cellular structures [36,37,38,39]. Thus, the groups of eukaryotic proteins found in this work, defined as ribosomal S1 proteins (chloroplast and mitochondrial), as well as Ts elongation factors (chloroplast and mitochondrial) and analysis of their individual S1 domains (number, identity), can be considered another proof of the theory symbiogenesis.

## 3. Materials and Methods

### 3.1. Distribution of Proteins Containing the S1 Domain Among the Three Domains of Life

To obtain data on the distribution of bacterial, eukaryotic and archaeal proteins containing the S1 domain, all available records were collected in the UniProt database version 2022_01 (https://prosite.expasy.org/, accessed on 1 October 2024) using the query “profile: prosite ps50126” (PS50126—domain profile identifier S1 in PROSITE [64]).

### 3.2. Number of Structural Repeats in Proteins

The number of structural S1 domains in proteins was collected for the corresponding records from the SMART protein domain database (http://smart.embl-heidelberg.de/, accessed on 1 October 2024) [65].

### 3.3. Realization

The realization of data search and collection, analysis and presentation of data were performed using the high-level programming language Python 3 (https://www.python.org/, accessed on 1 October 2024).

### 3.4. Alignment and Analysis of Sequences

Multiple sequence alignment was implemented using Clustal Omega (https://www.ebi.ac.uk/Tools/msa/clustalo/, accessed on 1 October 2024), and standard parameters were used (Appendix A). Phylogenetic trees were built by MEGA (https://www.megasoftware.net, accessed on 14 November 2024) using a Neighbor Joining Method. 

### 3.5. Taxonomic Diversity

The organisms were classified into major taxonomic categories according to the NCBI taxonomic database (http://www.ncbi.nlm.nih.gov/taxonomy, accessed on 1 October 2024).

## 4. Conclusions

The S1 domain, as a structural domain, can be found in a wide variety of proteins and protein complexes in bacteria, archaea and eukaryotes. The main function of the S1 domain is the binding of nucleic acids. The S1 domain can be present in a different number of copies in different proteins, while its 3D structure remains conserved. The study in this work of all records containing the S1 domain available in the UniProt protein sequence database made it possible to prove that the number and features of the S1 domain in bacterial, archaeal and eukaryotic proteins reflects the identified characteristics for other structural repeats; namely, the conservatism of close repeats and the predominance of the number of repetitions is a multiple of two. For the first time, a group of ChRpS1 with characteristic properties of bacterial S1 ribosomal proteins from the Cyanobacteria has been identified. Also, we have identified chloroplast and mitochondrial Ts elongation factors containing two structural S1 domains in a separate group. The data obtained about individual S1 domains of Cyanobacteria and Alphaproteobacteria allowed us to consider the RNA-binding S1 domain as one of the evolutionary markers of symbiogenesis.

## Figures and Tables

**Figure 1 ijms-25-13057-f001:**
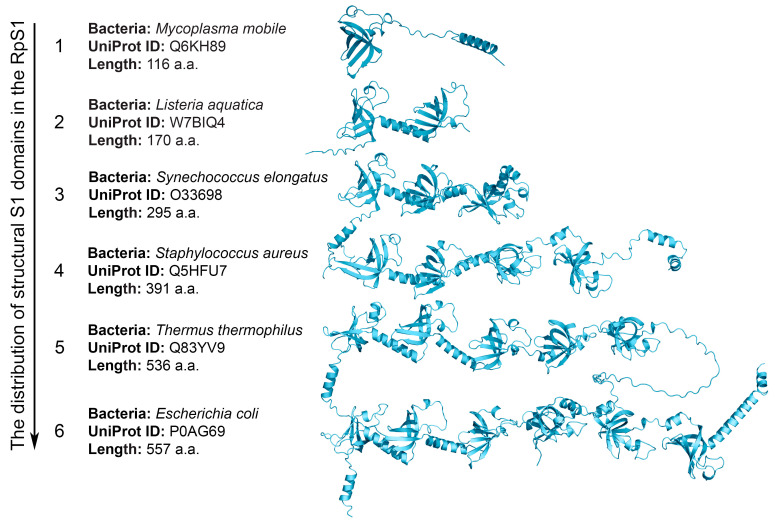
Schematic representation of the distribution of structural S1 domains in the RpS1 family in different bacterial phyla. Three-dimensional structures are predicted by the AlphaFold program (https://alphafold.ebi.ac.uk/, accessed on 1 October 2024).

**Figure 2 ijms-25-13057-f002:**
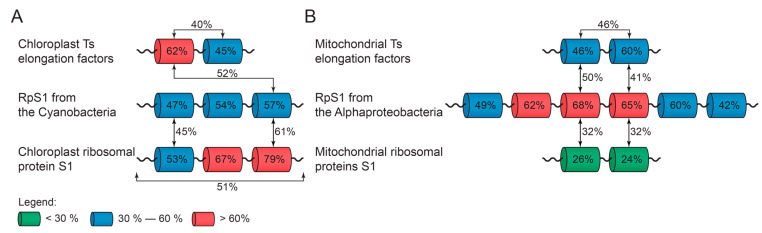
Percentage of identity within each S1 domain and between domains in RpS1 (Cyanobacteria, Alphaproteobacteria) in comparison with groups of chloroplasts (**A**) and mitochondrial (**B**) Ts elongation factors. The domains with the highest sequence identity are marked. The identity percentages between such domains are shown. The identity percentage reflects the results of multiple alignments of individual S1 domains.

**Figure 3 ijms-25-13057-f003:**
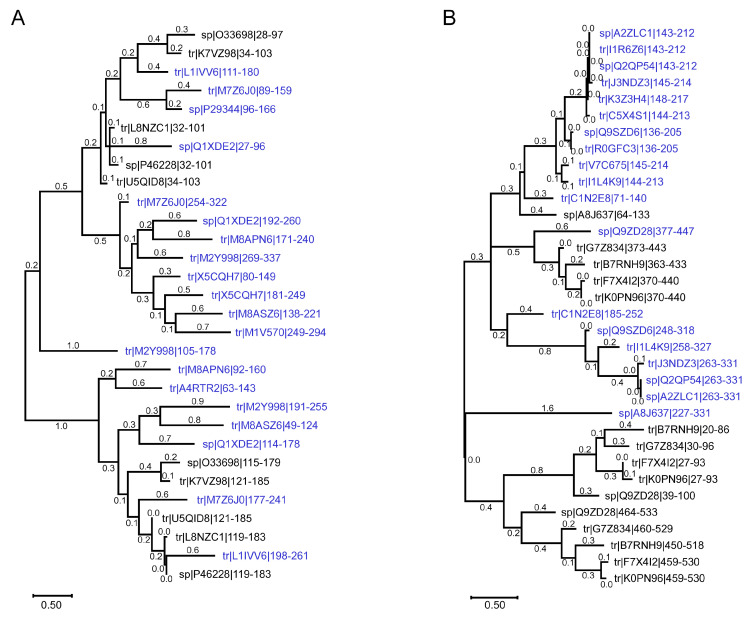
(**A**). Phylogenetic tree of S1 domain sequences of chloroplast eukaryotic proteins (blue) with the S1 domain and some three-domain-containing RpS1 of the Cyanobacteria (**B**). Phylogenetic tree of S1 domain sequences of mitochondrial elongation factor Ts (blue) with the S1 domain and some six-domain-containing RpS1 of the Alphaproteobacteria. Positions of the S1 domain are marked. Bar: 0.5 substitutions per nucleotide position.

**Figure 4 ijms-25-13057-f004:**
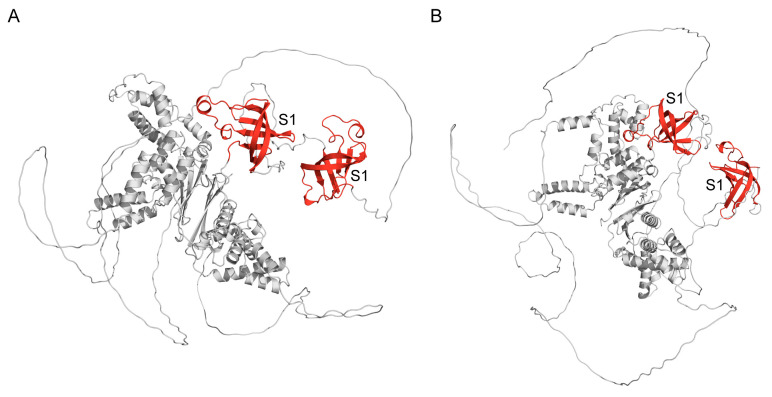
Structural models (obtained by AlphaFold) of chloroplastic elongation factor Ts (**A**) from *A. thaliana* (UniProt ID: Q9SZD6) and mtEF-Ts (**B**) from *B. distachyon* (UniProt ID: I1HXX8). S1 domains are highlighted in red.

**Figure 5 ijms-25-13057-f005:**
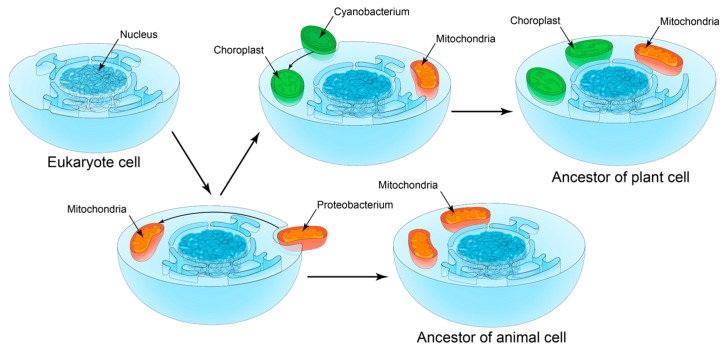
Schematic representation of possible evolution paths in the symbiogenesis theory. Figure is licensed under CC BY-SA 3.0 and adapted from https://en.wikipedia.org/wiki/Symbiogenesis#/media/File:Serial_endosymbiosis.svg (accessed on 1 October 2024).

**Table 1 ijms-25-13057-t001:** Distribution of the S1 domain in eukaryotic, archaeal and bacterial proteins.

Domain of Life	Number of Sequences Containing S1 Domain (Total Sequences)	Number of S1 Domains	Size of Proteins Containing the S1 Domain, a.a.r.	Name of the Shortest Protein (UniProt ID)	Name of the Longest Protein (UniProt ID)
Archaea	1162 (1162)	1	79–896	Translation initiation factor IF-2 subunit alpha (D2EE74)	Archaea-specific RecJ-like exonuclease, contains DnaJ-type Zn finger domain protein (V4XD76)
Bacteria	19,137 (24,142)	1	36–1596	RNA-binding S1 domain-containing protein (F3MV04)	Rne/Rng family ribonuclease (S3BLZ2)
	354 (24,142)	2	113–2058	Ribosomal protein S1 (F8XLB3)	Uncharacterized protein (W2CTN3)
	449 (24,142)	3	183–1864	30S ribosomal protein S1 (S2RAG1)	Uncharacterized protein (I9IS49)
	1777 (24,142)	4	279–850	30S ribosomal protein S1 (U5U0V0)	4-hydroxy-3-methylbut-2-enyl diphosphate reductase (F9VK01)
	418 (24,142)	5	378–1008	30S ribosomal protein S1 (P14128)	RNA-binding S1 domain protein (F0S7A1)
	2002 (24,142)	6	485–932	30S ribosomal protein S1 (Q4ECF4)	30S ribosomal protein S1 (B0VI92)
	4 (24,142)	7	600	RNA-binding S1 domain protein (C7NE91)	
	1 (24,142)	8	681	RNA-binding S1 domain protein (D1AGH2)	
Eukaryota	2486 (3263)	1	52–3347	Eukaryotic translation initiation factor 2 subunit 1 (P83268)	Ubiquitinyl hydrolase 1 (T1J6G6)
	247 (3263)	2	99–2273	Chloroplast ribosomal protein S1 (A1BQM4)	RNA helicase (T1IZH9)
	111 (3263)	3	263–3075	30S ribosomal protein S1, chloroplastic (Q1XDE2)	Pre-rRNA processing protein (F0VM47)
	22 (3263)	4	341–2871	Uncharacterized protein (B9G786)	Pre-rRNA processing protein (S7W597)
	22 (3263)	5	812–2865	Uncharacterized protein (F0ZDF8)	Pre-rRNA processing protein (S8GF05)
	12 (3263)	6	561–2366	30S ribosomal protein S1 (M3TEP3)	Uncharacterized protein (K0SL12)
	10 (3263)	7	861–1835	Uncharacterized protein (I1BSM3)	Uncharacterized protein (R1E487)
	21 (3263)	8	771–2032	cDNA FLJ61218, highly similar to RRP5 protein homolog (B4DES7)	Programmed cell death 11, putative (G0QXP6)
	18 (3263)	9	1210–2130	rRNA biogenesis protein RRP5 (H1VL07)	Protein RRP5-like protein (S9WVU3)
	49 (3263)	10	1427–2437	LOC779090 protein (A0JMT9)	Protein RRP5 like protein (M7BPG8)
	135 (3263)	11	1300–2245	Part of small ribosomal subunit processosome (Contains u3 snorna) (I2K1K0)	Rrna biogenesis protein rrp5 (W7TCI2)
	99 (3263)	12	1431–2229	Protein RRP5 homolog (H9EZV6)	Predicted protein (C1NA95)
	17 (3263)	13	1592–2018	Uncharacterized protein (L1JFG5)	Uncharacterized protein (C1ECK5)
	12 (3263)	14	1789–2077	Uncharacterized protein (R0G309)	Uncharacterized protein (K8EKT3)
	2 (3263)	15	1869–2384	Uncharacterized protein (A4S384)	(spotted green pufferfish) hypothetical protein (Q4S7N9)

**Table 2 ijms-25-13057-t002:** Chloroplast and mitochondrial eukaryotic proteins with the S1 domain.

UniProt ID	Protein Name	Protein Size, a.a.r.	Number of S1 Domains	Source	Phylum, Class
	Ribosomal protein S1				
X5CQH7	30S plastidal ribosomal protein S1	329	2	*Tisochrysis lutea*	Haptophyta
M2Y998	30S ribosomal protein S1 (Plastid)	391	3	*Galdieria sulphuraria*	Rhodophyta
M2XXH5	30S ribosomal protein S1 (Plastid)	523	3	*Galdieria sulphuraria*	Rhodophyta
M8CHE3	30S ribosomal protein S1, chloroplastic	155	1	*Aegilops tauschii*	Streptophyta
M7Z587	30S ribosomal protein S1, chloroplastic	155	1	*Triticum urartu*	Streptophyta
Q1XDE2	30S ribosomal protein S1, chloroplastic	263	3	*Neopyropia yezoensis*	Rhodophyta
P51345	30S ribosomal protein S1, chloroplastic	263	3	*Porphyra purpurea*	Rhodophyta
R7WGF8	30S ribosomal protein S1, chloroplastic	268	1	*Aegilops tauschii*	Streptophyta
M8APN6	30S ribosomal protein S1, chloroplastic	324	2	*Triticum urartu*	Streptophyta
M8ASZ6	30S ribosomal protein S1, chloroplastic	343	2	*Triticum urartu*	Streptophyta
M8CDW9	30S ribosomal protein S1, chloroplastic	398	3	*Aegilops tauschii*	Streptophyta
M7Z6J0	30S ribosomal protein S1, chloroplastic	398	3	*Triticum urartu*	Streptophyta
P29344	30S ribosomal protein S1, chloroplastic	411	3	*Spinacia oleracea*	Streptophyta
M0ZL57	30S ribosomal protein S1, chloroplastic	411	3	*Solanum tuberosum*	Streptophyta
M1A029	30S ribosomal protein S1, chloroplastic	415	3	*Solanum tuberosum*	Streptophyta
Q93VC7	30S ribosomal protein S1, chloroplastic	416	3	*Arabidopsis thaliana*	Streptophyta
W0RYL6	Chloroplast 30S ribosomal protein S1	263	3	*Porphyridium purpureum*	Rhodophyta
M1VII4	Chloroplast ribosomal protein S1	447	3	*Cyanidioschyzon merolae*	Rhodophyta
L1IVV6	Ribosomal protein S1, chloroplastic	404	3	*Guillardia theta*	Cryptophyceae
M1V570	Mitochondrial ribosomal protein S1	424	2	*Cyanidioschyzon merolae*	Rhodophyta
A4RTR2	Putative mitochondrial ribosomal protein S1	352	2	*Ostreococcus lucimarinus*	Chlorophyta
	**Elongation factor Ts**				
Q9SZD6	Elongation factor Ts, chloroplastic	953	2	*Arabidopsis thaliana*	Streptophyta
Q2QP54	Elongation factor Ts, chloroplastic	1123	2	*Oryza sativa subsp. japonica*	Streptophyta
A8J637	Elongation factor Ts, chloroplastic	1013	2	*Chlamydomonas reinhardtii*	Chlorophyta
A2ZLC1	Elongation factor Ts, chloroplastic	1123	2	*Oryza sativa subsp. indica*	Streptophyta
K3Z3H4	Elongation factor Ts, mitochondrial	988	2	*Setaria italica*	Streptophyta
I1HXX8	Elongation factor Ts, mitochondrial	962	2	*Brachypodium distachyon*	Streptophyta
C1N2E8	Elongation factor Ts, mitochondrial	844	2	*Micromonas pusilla*	Chlorophyta
A9SG13	Elongation factor Ts, mitochondrial	899	2	*Physcomitrium patens*	Streptophyta
W1NPM5	Elongation factor Ts, mitochondrial	1164	2	*Amborella trichopoda*	Streptophyta
R0GFC3	Elongation factor Ts, mitochondrial	953	2	*Capsella rubella*	Streptophyta
F6HH07	Elongation factor Ts, mitochondrial	1135	2	*Vitis vinifera*	Streptophyta
I1R6Z6	Elongation factor Ts, mitochondrial	1123	2	*Oryza glaberrima*	Streptophyta
B9RKL9	Elongation factor Ts, mitochondrial	972	2	*Ricinus communis*	Streptophyta
M4D3M7	Elongation factor Ts, mitochondrial	1770	2	*Brassica rapa subsp. pekinensis*	Streptophyta
I1L4K9	Elongation factor Ts, mitochondrial	1135	2	*Glycine max*	Streptophyta
R7W2H4	Elongation factor Ts, mitochondrial	937	2	*Aegilops tauschii*	Streptophyta
M5VUX8	Elongation factor Ts, mitochondrial	1010	2	*Prunus persic*	Streptophyta
J3NDZ3	Elongation factor Ts, mitochondrial	1200	2	*Oryza brachyantha*	Streptophyta
C1DZB9	Elongation factor Ts, mitochondrial	876	2	*Micromonas commoda*	Chlorophyta
G7KPU0	Elongation factor Ts, mitochondrial	1054	2	*Medicago truncatula*	Streptophyta
C5X4S1	Elongation factor Ts, mitochondrial	937	2	*Sorghum bicolor*	Streptophyta
K8EFG1	Elongation factor Ts, mitochondrial	831	2	*Bathycoccus prasinos*	Chlorophyta
M0ZU44	Elongation factor Ts, mitochondrial	1050	2	*Solanum tuberosum*	Streptophyta
V7C675	Elongation factor Ts, mitochondrial	1134	2	*Phaseolus vulgaris*	Streptophyta
I1J7Z7	Elongation factor Ts, mitochondrial	1133	2	*Glycine max*	Streptophyta
V4MGB9	Elongation factor Ts, mitochondrial	979	2	*Eutrema salsugineum*	Streptophyta
M7YWW3	Elongation factor Ts, mitochondrial	987	2	*Triticum urartu*	Streptophyta
V4UQR1	Elongation factor Ts, mitochondrial	902	2	*Citrus clementina*	Streptophyta
W9SL34	Elongation factor Ts, mitochondrial	1060	2	*Morus notabilis*	Streptophyta

## Data Availability

The data are contained within the article and Appendix A.

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
