# Peer review of "RNA-Binding S1 Domain in Bacterial, Archaeal and Eukaryotic Proteins as One of the Evolutionary Markers of Symbiogenesis"

_ijms, 2024, doi:10.3390/ijms252313057_

Round 1

Reviewer 1 Report

Comments and Suggestions for Authors

In the manuscript " RNA-binding S1 Domain in Bacterial, Archaeal and Eukaryotic Proteins as One of the Evolutionary Markers of Symbiogenesis”, Deryusheva et al made an attempt to claim S1 capable of being symbiogenesis marker. The authors proposed this on the basis of sequence and structure. This manuscript is of scientific and practical interest, and relatively well structured and written. However, given that OB fold proteins are many, is S1 domain actually the one marker of symbiogenesis should be further studied. Is S1 domain present at earliest stage during evolution? Which organism(s) does not contain this RNA-binding S1 domain should be indicated. If being evolutionary marker, their partner protein(s) must be co-evolved, and, thus, does the RNA-binding S1 domain have this partner(s) as co-marker? The cited papers should be updated including published review articles. Other evolutionary markers of symbiogenesis should be compared and discussed. Structural identity (e.g. rmsd?) among these proteins should be given. Overall, this manuscript is worth of publication in ijms through minor revision.

Author Response

In the manuscript " RNA-binding S1 Domain in Bacterial, Archaeal and Eukaryotic Proteins as One of the Evolutionary Markers of Symbiogenesis”, Deryusheva et al made an attempt to claim S1 capable of being symbiogenesis marker. The authors proposed this on the basis of sequence and structure. This manuscript is of scientific and practical interest, and relatively well structured and written. However, given that OB fold proteins are many, is S1 domain actually the one marker of symbiogenesis should be further studied.

Answer:

Thank you for reviewing our article.

Is S1 domain present at earliest stage during evolution?

Answer:

As we mentioned in the article, the OB fold, to which the S1 domain belongs, “is one of the “oldest” protein folds” [doi.org/10.1016/S0092-8674(00)81844-9, doi:10.1002/j.1460-2075.1993.tb05726.x, doi.org/10.2174/1389203033487207]. Moreover, the S1 domain is part of the 30S ribosome subunit and can be found even in bacteria with a small genome size [doi:10.1002/prot.26084]. So, without a doubt, S1 domain present at earliest stage during evolution.

Which organism(s) does not contain this RNA-binding S1 domain should be indicated.

Answer:

It can be assumed that the S1 domain in various proteins can be found in any organism. In bacteria, it is part of the 30S ribosomal subunit, and in archaea and eukaryotes (for sequences available in Uniprot), it is part of proteins necessary for functioning. It may be absent in archaeal organisms living in extreme conditions, with their own specific adaptations.

If being evolutionary marker, their partner protein(s) must be co-evolved, and, thus, does the RNA-binding S1 domain have this partner(s) as co-marker?

Answer:

Translation initiation and elongation factors, which include S1 domains, are themselves considered molecular markers of symbiogenesis [PMID: 22135192, PMID: 15494441]. Our GO analysis of S1 domains in ribosomal S1 proteins revealed that bacterial protein sequences in S1 mainly have 3 types of molecular functions: RNA binding activity, nucleic acid activity and ribosome structural constituent  [PMID: 38537772]. As the part of 30S ribosomal subunit ribosomal protein S1 cooperates with other ribosomal proteins (S2, S3, S6, and S18) to form a dynamic mesh near the mRNA exit and entrance channels to modulate the binding, folding and movement of mRNAS1 domains formed 30S ribosomal subunit [PMID: 30177741]. So, the authors are not aware of any studies on the classification of other ribosomal proteins (S1 domain partner(s)) as markers.

The cited papers should be updated including published review articles.

Answer:

References were updated by published review articles.

Other evolutionary markers of symbiogenesis should be compared and discussed. Structural identity (e.g. rmsd?) among these proteins should be given.

Answer:

Other evolutionary markers of symbiogenesis were indicated in Discussion.

Overall, this manuscript is worth of publication in ijms through minor revision.

Answer:

The authors thank the reviewer for his comments and suggestions for improving our manuscript.

Reviewer 2 Report

Comments and Suggestions for Authors

In the study titled “RNA-binding S1 Domain in Bacterial, Archaeal, and Eukaryotic Proteins as an Evolutionary Marker of Symbiogenesis,” Evgenia and colleagues conducted a comprehensive analysis of proteins across bacterial, eukaryotic, and archaeal domains to map the distribution of the S1 domain. Their research highlighted a subset of eukaryotic chloroplast S1 ribosomal proteins (ChRpS1) that share characteristics with bacterial S1 ribosomal proteins (RpS1) from Cyanobacteria. Additionally, they identified a distinct group comprised of chloroplast and mitochondrial elongation factors Ts, each containing dual S1 structural domains. Comparative analysis of these mitochondrial elongation factors Ts revealed specific similarities and differences with RpS1 proteins from the Cyanobacteria phylum and Alphaproteobacteria class, suggesting the S1 domain’s potential role as an evolutionary indicator of symbiogenesis between bacterial and eukaryotic organisms.

However, the paper primarily cataloged proteins by name, number, and size, lacking in-depth analysis which limits its contribution to existing knowledge. 

Comments:

1. Incorporate evolutionary tree analyses to better elucidate the relationships and evolutionary trajectories of these proteins.

2. Perform a sequence analysis of the conserved S1 domain to deepen the understanding of its structural and functional roles.

3. Clearly articulate the connection between the S1 domain in elongation factor Ts and its evolutionary significance.

4. Address typographical errors such as the redundant phrase in line 59, “OB-fold fold.

Author Response

Comments and Suggestions for Authors

In the study titled “RNA-binding S1 Domain in Bacterial, Archaeal, and Eukaryotic Proteins as an Evolutionary Marker of Symbiogenesis,” Evgenia and colleagues conducted a comprehensive analysis of proteins across bacterial, eukaryotic, and archaeal domains to map the distribution of the S1 domain. Their research highlighted a subset of eukaryotic chloroplast S1 ribosomal proteins (ChRpS1) that share characteristics with bacterial S1 ribosomal proteins (RpS1) from Cyanobacteria. Additionally, they identified a distinct group comprised of chloroplast and mitochondrial elongation factors Ts, each containing dual S1 structural domains. Comparative analysis of these mitochondrial elongation factors Ts revealed specific similarities and differences with RpS1 proteins from the Cyanobacteria phylum and Alphaproteobacteria class, suggesting the S1 domain’s potential role as an evolutionary indicator of symbiogenesis between bacterial and eukaryotic organisms. However, the paper primarily cataloged proteins by name, number, and size, lacking in-depth analysis which limits its contribution to existing knowledge. 

Answer:

The authors thank the reviewer for his comments and suggestions for improving our manuscript.

Comments:

  1. Incorporate evolutionary tree analyses to better elucidate the relationships and evolutionary trajectories of these proteins.

Answer:

At the suggestion of the reviewer, we have added the analyses of the evolutionary trees of the studied sequences in a separate File S2 and the manuscript.

  1. Perform a sequence analysis of the conserved S1 domain to deepen the understanding of its structural and functional roles.

Answer:

We have carried out a detailed analysis of the S1 sequences and its functional roles in our previous works [Proteins 2017, 85, 602–613, doi:10.1002/prot.25237; Molecules 2019, 24, 3681, doi:10.3390/molecules24203681; PLoS One 2019, 14, e0221370, doi:10.1371/journal.pone.0221370; Proteins 2021, 89, 1111–1124, doi:10.1002/prot.26084], some of the data are posted in the public domain on our website http://oka.protres.ru:4200. In this paper, we compared a group of chloroplast S1 ribosomal proteins (ChRpS1) and chloroplast and mitochondrial elongation factors Ts with bacterial S1 ribosomal proteins (RpS1) from the Cyanobacteria and Alphaproteobacteria to find a correlation between them. At the suggestion of the reviewer, we have added the alignment of the studied sequences to a separate File S2 and added a link to it in the text of the article.  

  1. Clearly articulate the connection between the S1 domain in elongation factor Ts and its evolutionary significance.

Answer:

We expanded in the text the description of the structure and connection between the S1 domain in elongation factor Ts and its evolutionary significance.

  1. Address typographical errors such as the redundant phrase in line 59, “OB-fold fold.

Answer:

We have done.
